# GENERAL NEURAL GAUGE FIELDS

**Fangneng Zhan, Lingjie Liu, Adam Kortylewski, Christian Theobalt**
Max Planck Institute for Informatics, 66123, Germany
{fzhan,lliu,akortyle,theobalt}@mpi-inf.mpg.de

## ABSTRACT

The recent advance of neural fields, such as neural radiance fields, has significantly pushed the boundary of scene representation learning. Aiming to boost the computation efficiency and rendering quality of 3D scenes, a popular line of research maps the 3D coordinate system to another measuring system, e.g., 2D manifolds and hash tables, for modeling neural fields. The conversion of coordinate systems can be typically dubbed as *gauge transformation*, which is usually a pre-defined mapping function, e.g., orthogonal projection or spatial hash function. This begs a question: can we directly learn a desired gauge transformation along with the neural field in an end-to-end manner? In this work, we extend this problem to a general paradigm with a taxonomy of discrete & continuous cases, and develop an end-to-end learning framework to jointly optimize the gauge transformation and neural fields. To counter the problem that the learning of gauge transformations can collapse easily, we derive a general regularization mechanism from the principle of information conservation during the gauge transformation. To circumvent the high computation cost in gauge learning with regularization, we directly derive an information-invariant gauge transformation which allows to preserve scene information inherently and yield superior performance.

## 1 INTRODUCTION

Representing 3D scenes with high efficiency and quality has been a long-standing target in computer vision and computer graphics research. Recently, the implicit representation of neural radiance fields (Mildenhall et al., 2021) has shown that a 3D scene can be modeled with neural networks, which achieves compelling visual quality and low memory footprint. However, it suffers from long training times. The explicit voxel-based methods (Yu et al., 2021a; Sun et al., 2022) emerged with faster convergence but higher memory requirements, due to the use of the 3D voxel grids as scene representation. To strike a good balance between computation efficiency and rendering quality, EG3D (Chan et al., 2022) proposed to project the 3D coordinate system to a tri-plane system. Along with this line of research, TensoRF (Chen et al., 2022) factorizes 3D space into compact low-rank tensors; Instant-NGP (Müller et al., 2022) models the 3D space with multi-resolution hash grids to enable remarkably fast convergence speed.

These recent works (e.g., EG3D and Instant-NGP) share the same prevailing paradigm by converting the 3D coordinate of neural fields to another coordinate system. Particularly, a coordinate system of the scene (e.g., 3D coordinate and hash table) can be regarded as a kind of gauge, and the conversion between the coordinate systems can be referred to as **gauge transformation** (Moriyasu, 1983). Notably, existing gauge transformations in neural fields are usually pre-defined functions (e.g., orthogonal mappings and spatial hash functions (Teschner et al., 2003)), which are sub-optimal for modeling the neural fields as shown in Fig. 1. In this end, a learnable gauge transformation is more favored as it can be optimized towards the final objective. This raises an essential question: how to learn the gauge transformations along with the neural fields. Some previous works explore a special case of this problem, e.g., NeuTex (Xiang et al., 2021) and NeP (Ma et al., 2022) aim to transform 3D points into continuous 2D manifolds. However, a general and unified learning paradigm for various gauge transformations has not been established or explored currently.

In this work, we introduce general **Neural Gauge Fields** which unify various gauge transformations in neural fields with a special focus on how to learn the gauge transformations along with neural fields. Basically, a gauge is defined by gauge parameters and gauge basis, e.g., codebook indices

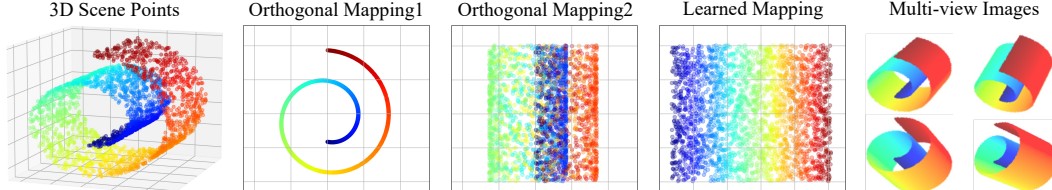

Figure 1: Conceptual illustration of a gauge transformation from 3D point coordinates to a 2D plane. Instead of naively employ a pre-defined orthogonal mapping which incurs overlap on the 2D plane, the proposed neural gauge fields aim to learn the mapping along with neural fields driven by multi-view synthesis loss.

and codebook vectors, which can be continuous or discrete. Thus, the gauge transformations for neural fields can be duly classified into continuous cases (e.g., triplane space) and discrete cases (e.g., a hash codebook space). We then develop general learning paradigms for continuous cases and discrete cases, which map a 3D point to a continuous coordinate or a discrete index in the target gauge, respectively.

As typical cases, we study continuous mapping from 3D space to the 2D plane and discrete mapping from 3D space to 256 discrete vectors. As shown in Fig. 2 and 3, we observed that naively optimizing the gauge transformations with the neural fields severely suffers from gauge collapse, which means the gauge transformations will collapse to a small region in a continuous case or collapse to a small number of indices in a discrete case Baevski et al. (2019); Kaiser et al. (2018). To regularize the learning of gauge transformation, a cycle consistency loss has been explored in Xiang et al. (2021); Ma et al. (2022) to avoid many-to-one mapping; a structural regularization is also adopted in Tretschk et al. (2021) to preserve local structure by only predicting a coordinate offset. However, the cycle consistency regularization tends to be heuristic without grounded derivation while the structural regularization is constrained to regularize continuous cases. In this work, we introduce a more intuitive **Info**rmation **Reg**ularization (**InfoReg**) from the principle of information conservation during gauge transformation. By maximizing the mutual information between gauge parameters, we successfully derive general regularization forms for gauge transformations. Notably, a geometric uniform distribution and a discrete uniform distribution are assumed as the prior distribution of continuous and discrete gauge transformations, respectively, where an Earth Mover's distance and a KL divergence are duly applied to measure the distribution discrepancy for regularization.

Learning the gauge transformation with regularization usually incurs high computation cost which is infeasible for some practical applications such as fast scene representation. In line with relative information conservation, we directly derive an **Info**rmation-**Inv**ariant (**InfoInv**) gauge transformation which allows to preserve scene information inherently in gauge transformations and obviate the need for regularizations. Particularly, the derived InfoInv coincides with the form of position encoding adopted in NeRF (Mildenhall et al., 2021), which provides certain rationale for the effectiveness of position encoding in neural fields.

The contributions of this work are threefold. First, we develop a general framework of neural gauge fields which unifies various gauge transformations in neural fields, and gives general learning forms for continuous and discrete gauge transformations. Second, we strictly derive a regularization mechanism for gauge transformation learning from the perspective of information conservation during gauge transformation that outperform earlier heuristic approaches. Third, we present an information nested function to preserve scene information inherently during gauge transformations.

## 2 RELATED WORK

Recent work has demonstrated the potential of neural radiance fields (Mildenhall et al., 2021) and its extensions for multifarious vision and graphics applications, including fast view synthesis (Liu et al., 2020; Yu et al., 2021b; Hedman et al., 2021; Lindell et al., 2021; Neff et al., 2021; Yu et al., 2021a; Reiser et al., 2021; Sun et al., 2022), generative models (Schwarz et al., 2020; Niemeyer & Geiger, 2021; Gu et al., 2022; Chan et al., 2021; Or-El et al., 2022), surface reconstruction (Wang et al., 2021; Oechsle et al., 2021; Yariv et al., 2021), etc. Under this context, various gauge transformations have

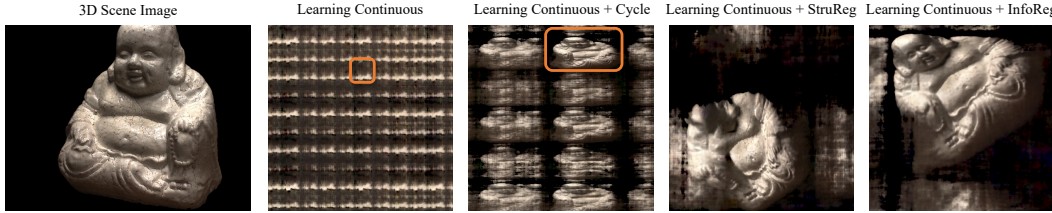

Figure 2: Continuous gauge transformation from 3D scene to 2D square. Without regularization, the learned gauge transformation will collapse to a small region in the 2D square. Including cycle regularization (Xiang et al., 2021) or structural regularization (StruReg) Tretschk et al. (2021) don't fully solve the problem, and our InfoReg achieves a clearly better regularization performance.

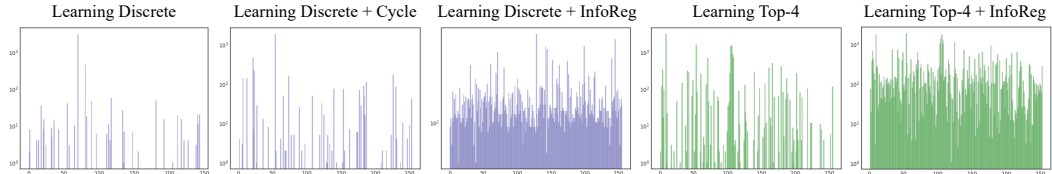

Figure 3: Discrete gauge transformation from 3D scene to 256 discrete vectors. Without regularization, the learned gauge transformation will collapse to a small number of vector indices (horizontal axis). The learning with cycle regularization (Xiang et al., 2021) is still significantly collapse-prone, while our InfoReg enables to alleviates the collapse substantially.

been intensely explored in neural fields for different purposes, e.g., reducing computation cost (Chan et al., 2022), fast convergence (Chen et al., 2022; Müller et al., 2022), editable UV map (Xiang et al., 2021; Ma et al., 2022).

A pre-defined mapping function is usually adopted as gauge transformation. A simple case is orthogonal mapping, which projects 3D space into a continuous low-dimension space. For instance, EG3D (Chan et al., 2022; Peng et al., 2020) orthogonally projects 3D space into single plane or triplane space for scene modelling. TensoRF (Chen et al., 2022) further extends this line of research for fast and efficient optimization by mapping the 3D scene space into a set of low-dimension tensors. With a pre-defined spatial hash function, the recent Instant-NGP (Müller et al., 2022) maps 3D grid points into hash table indices and achieves a speedup of several orders of magnitude. Besides, PREF (Huang et al., 2022) explores to transform 3D scene points to a set of Fourier basis via Fourier transformation, which helps to reduce the aliasing caused by interpolation.

Some works also explore learning the gauge transformation for downstream tasks in neural fields. For example, NeuTex (Xiang et al., 2021) and NeP (Ma et al., 2022) propose to learn the mapping from 3D points to 2D texture spaces; GRAM (Deng et al., 2022) learns to map the 3D space into a set of implicit 2D surfaces; VQ-AD Takikawa et al. (2022) enables compressive scene representation by mapping coordinates to discrete codebook, which is a discrete case of gauge transformation. However, all above works suffer from gauge collapse without regularization or designed initialization. Besides, a popular line of research (Tretschk et al., 2021; Tewari et al., 2022; Park et al., 2020; Pumarola et al., 2020; Liu et al., 2021; Peng et al., 2021) learns the deformation from the actual 3D space into another (canonical) 3D space, where a structural regularization is usually applied by predicting a deformation offset instead of the absolute coordinate.

## 3 METHODS

This section will start with the introduction of neural gauge fields framework, followed by the derivation of learning regularization based on information conservation. The nested gauge transformation which aims to preserve scene information inherently is then introduced and derived. Some discussion and insights for the application of gauge transformations are drawn in the end.

### 3.1 Neural Gauge Fields Framework

In normal usage, a gauge defines a measuring system, e.g., pressure gauge and temperature gauge. Under the context of neural fields, a measuring system (i.e., gauge) is a specification of parameters to index a neural field, e.g., 3D Cartesian coordinate system, triplane in EG3D (Chan et al., 2022), hash table in Instant-NGP (Müller et al., 2022). The transformation between different measuring systems is dubbed as **Gauge Transformation**.

Based on this intuition, we introduce the general framework of **Neural Gauge Fields** which consists of a gauge transformation and a neural field. A neural gauge field aims to transform the original space to another gauge system to index a neural field. This additional transform could introduce certain bonus to the neural field, e.g., low memory cost, high rendering quality, or explicit texture, depending on the purpose of the model. The gauge transformation could be a pre-defined function e.g., an orthogonal mapping in EG3D and a spatial hash function in Instant-NGP, or directly learned by the neural network along with neural fields. Intuitively, compared with a pre-defined function, a learnable gauge transformation is more favorable as it can be optimized towards the final use case of the neural field and possibly yields better performance. For learning neural gauge fields, we disambiguate between two cases: continuous (e.g., 2D plane and sphere surface) and discrete (e.g., hash table space) mappings.

**Continuous Gauge Transformation.** As the target gauge system is continuous, the gauge transformation can be handled by directly regressing the coordinate (i.e., gauge parameters) in the target gauge. For a point $x \in \mathbb{R}^3$ in the 3D space and a target gauge, a gauge network $\mathcal{M}$ (a MLP structure or transformation matrix) can be employed to predict the target coordinate as $x' = \mathcal{M}(x), x' \in \mathbb{R}^N$, where $N$ is the coordinate dimension of the target gauge. Then, the predicted coordinate $x'$ can be fed into an MLP or used to look up feature vectors to model neural fields.

**Discrete Gauge Transformation.** Without loss of generality, we assume there is a codebook with $N$ discrete vectors $[v_1, v_2, \cdots, v_N]$, and the discrete gauge transformation aims to transform a 3D point to a discrete codebook index (i.e., gauge parameter). In line with the setting of Instant-NGP, we divide the 3D space into a grid of size $M * M * M$. For each grid point $x$, we apply a gauge network $\mathcal{M}$ to predict its discrete distribution $P_x \in \mathbb{R}^N$ over the $N$ discrete vectors, followed by a Top-k (k=1 by default) operation to select the Top-k probable indices $i_1, i_2, \cdots, i_k$ as below:

$$P_x = \mathcal{M}(x), \quad i_1, i_2, \cdots, i_k = \text{Top-k}(P_x) \qquad P_x \in \mathbb{R}^N. \tag{1}$$

The produced indices are usually used to lookup corresponding vectors from the codebook [1], and the features of other 3D points can be acquired by interpolating the nearby grid point vectors. As the Argmax in Top-k operation is not differentiable, a reparameterization trick (Bengio et al., 2013) is applied to approximate gradient by replacing Argmax with Softmax in back-propagation. The pseudo code of the forward & backward propagation of discrete cases is given in Algorithm 1.

We now have established the general learning paradigms for continuous and discrete gauge transformations. In our setting, the gauge transformations are optimized along with the neural field supervised by the per-pixel color loss between the rendering and the ground-truth multi-view images. However, we found the learning of gauge transformations is easily stuck in local minimal. Specifically, the gauge transformation tends to collapse to a small region in continuous cases as in Fig. 2 or collapse to a small number of indices in discrete cases as in Fig. 3, highlighting the need for an additional regularization to enable an effective learning of continuous and discrete gauge transformations, which we will introduce in the following.

---

**Algorithm 1** Pseudo code of forward & backward propagation in learning discrete gauge transformation.

**Input:** A 3D point $x$, predicted distribution $P_x = [p_1, p_2, \cdots, p_N]$, codebook $\mathcal{V}$.
    **Forward propagation:**
    1. index = Argmax($P_x$)
    2. index_hard = One_Hot(index)
    3. feature lookup: $v_x$ = Matmul(index_hard, $\mathcal{V}$)
    **Backward propagation:**
    1. index_soft = Softmax($P_x$)
    2. feature lookup: $v_x$ = Matmul(index_soft, $\mathcal{V}$)
**Output:** feature $v_x$.

---

[1] Actually, the predicted indices can also be fed into an MLP for embedding, but this manner tends to be meaningless in practical applications and is seldomly adopted.

### 3.2 REGULARIZATION FOR LEARNING GAUGE TRANSFORMATIONS

Some previous works (Xiang et al., 2021; Ma et al., 2022) leverage a cycle consistency loss to regularize the mapping, preventing many-to-one mapping (gauge collapse) of 3D points. Alternatively, a structural regularization is also employed in some works (Tretschk et al., 2021; Tewari et al., 2022) to preserve local structure by predicting a residual offset instead of the absolute coordinate.

However, the cycle loss only works for certain continuous cases and shows poor performance for the regularization of discrete cases, while the structural regularization cannot be applied to discrete cases at all. In this work, we strictly derive a general regularization mechanism for gauge transformation from the perspective of information conservation which is superior to the heuristic regularizers used in prior work. Specifically, although the gauge of the scene has been changed, the intrinsic information of the scene is expected to be preserved during gauge transformation, which can be achieved by maximizing the mutual information between the original 3D points and target gauge parameters.

With the gauge network $\mathcal{M}$ for gauge transformation, the sets of original 3D points and target gauge parameters can be termed as $X$ and $Y = \mathcal{M}(X)$, respectively. Then, the mutual information between $X$ and $Y$ can be formulated as:

$$I(X, Y) = \iint p(x, y) \log \frac{p(x, y)}{p(x)p(y)} dx dy = \iint p(y|x)p(x) \log \frac{p(y|x)}{p(y)} dx dy, \qquad (2)$$

where $p(x)$ and $p(y)$ are the distributions of original 3D point $x$ and target gauge parameter $y$, $p(y|x)$ is the conditional distribution of $y$ produced by the gauge transformation of $x$. To avoid that the distribution of $Y$ collapses into a local region or small number of indices, we assume that $Y$ obeys a prior uniform distribution $q(y)$, and the KL divergence between $p(y)$ and $q(y)$ can be written as $KL(p(y)||q(y)) = \int p(y) \log \frac{p(y)}{q(y)} dy$. Then, the final regularization loss can translate into the following term (see detailed derivation in the Appendix):

$$\mathcal{L}_{reg} = \min_{p(y|x)} \left\{ -(\gamma + \epsilon) \cdot \mathbb{E}[KL(p(y|x)p(x)||p(y)p(x))] + \epsilon \cdot \mathbb{E}[KL(p(y|x)||q(y))] \right\}, \quad (3)$$

where $\gamma$ and $\epsilon$ denote the weight of regularization and prior distribution discrepancy.

Regarding the first term $KL(p(y|x)p(x)||p(y)p(x))$ in Eq. 3, we can replace the unbounded KL divergence with the upper-bounded JS divergence as: $JS(p(y|x)p(x)||p(y)p(x))$.

**Lemma 1** *Jensen-Shannon Mutual Information Estimator (Nowozin et al., 2016):*

$$JS(P, Q) = \max_T \left( \mathbb{E}_{x \sim p(x)}[-sp(-T(x))] - \mathbb{E}_{x \sim q(x)}[sp(T(x))] \right), \qquad (4)$$

where $sp(z) = log(1 + e^z)$. For $JS(p(y|x)p(x)||p(y)p(x))$, we thus have:

$$\max_T \left( \mathbb{E}_{(x^+, y) \sim p(x)p(y|x)}[-sp(-T(x^+, y))] - \mathbb{E}_{(x^-, y) \sim p(x)p(y)}[sp(T(x^-, y))] \right), \qquad (5)$$

where $T$ denotes a network which minimizes the distance of positive pairs and maximizes the distance of negative pairs. Please refer to section 4.1 for detailed implementation of $T$. Actually, CycleReg is special implementation of InfoReg by discarding the prior distribution discrepancy.

Now the first term of Eq. (3) has been well solved. We then turn to the second term $\mathbb{E}[KL(p(y|x)||q(y))]$ (i.e., the KL divergence between $p(y|x)$ and $q(y)$). This term can be solved according to the types of prior uniform distributions $q(y)$ assumed in continuous and discrete cases.

**Continuous Gauge Transformation.** To prevent that the gauge transformation collapses to a local region, we expect the prior distribution $q(y)$ is a uniform distribution over the target gauge space. Without loss of generality, we assume the target gauge space is a 2D plane as shown in Fig. 4 (a). Then, we uniformly sampled $h$ points on the 2D plane as a discrete approximation of the prior uniform distribution, which is denoted by $\overline{q}(y) = [u_1, u_2, \cdots, u_h], u_i \in \mathbb{R}^2$). With a set of 3D points transformed into 2D points, the conditional distribution $p(y|x)$ can also be approximated by sampling $h$ points from the predicted 2D points according to radiance contribution (see A.3), which is denoted by $\overline{p}(y|x) = [v_1, v_2, \cdots, v_h], v_i \in \mathbb{R}^2$. Considering $\overline{q}(y)$ and $\overline{p}(y|x)$ are distributions over a geometric space, KL divergence is inapposite to measure their discrepancy as the distribution geometry is not concerned in KL divergence. We thus resort to a geometry-aware metric: Earth Mover's Distance (EMD) (Rubner et al., 2000).

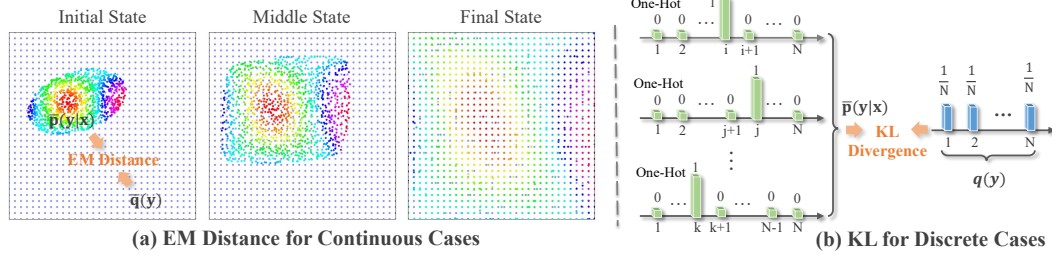

**(a) EM Distance for Continuous Cases**    **(b) KL for Discrete Cases**

Figure 4: (a) Earth Mover's distance to regularize the discrepancy between $\overline{p}(y|x)$ and $\overline{q}(y)$ in continuous case (2D plane), where $\overline{p}(y|x)$ and $\overline{q}(y)$ are predicted 2D points within a batch and uniformly sampled points in the 2D plane, respectively. (b) KL divergence to regularize the distribution discrepancy in discrete cases ($N$ basis vectors), where $\overline{p}(y|x)$ and $q(y)$ are the average of predicted one-hot distributions and a discrete uniform distribution, respectively.

Specifically, a cost matrix $C$ and transport matrix $M$ with size of $(h, h)$ can be defined where each entry $C_{ij}$ in $C$ gives the distance between points $v_i$ and $u_j$, each entry $M_{ij}$ in $M$ represents the amount of probability moved between points $v_i$ and $u_j$. Then, the EMD between $\overline{q}(y)$ and $\overline{p}(y|x)$ can be formulated as: $\mathcal{L}_{EMD} = \min_{M}(\sum_{i=1}^{h} \sum_{j=1}^{h} C_{ij} M_{ij})$. The distribution of predicted points will be regularized to fit the uniform distribution by minimizing $\mathcal{L}_{EMD}$ as shown in Fig. 4 (a).

**Discrete Gauge Transformation.** For discrete gauge transformation, we expect the prior distribution $q(y)$ to be a uniform distribution over the discrete vectors (i.e., the gauge basis), so that the target gauge parameters will not collapse to a small number of vector indices. Thus, the prior distribution $q(y)$ can be denoted by $q(y) = [\frac{1}{N}, \frac{1}{N}, \cdots, \frac{1}{N}]$ for N discrete vectors. As the one-hot distribution $p_i$ is predicted from 3D points as described in Algorithm 1, the conditional distribution $p(y|x)$ can be approximated by the average of all predicted one-hot distributions over a training batch, which is denoted by $\overline{p}(y|x) = \sum_{i=1}^{B} p_i/N$ where $B$ is the number of samples within a training batch. Then the KL divergence between $q(y)$ and $\overline{p}(y|x)$ can be duly computed as shown in Fig. 4 (b).

### 3.3 INFORMATION-INVARIANT GAUGE TRANSFORMATION

Learning gauge transformation and applying regularization to preserve information usually incur additional computation cost, which is not desired in many applications. We thus introduce an **Inf**ormation-**Inv**ariant (**InfoInv**) gauge transformation which enables to preserve scene information inherently and avoids the cumbersome process of learning regularization.

Specifically, for two positions $m$, $n$, we apply a gauge transformation $g$ to yield their new gauge parameters $f(m)$ and $f(n)$. As aiming to preserve the scene information, the relative position relationship between points should be preserved accordingly, i.e.:

$$Cos(g(m), g(n)) = \frac{< g(m), g(n) >}{\|g(m)\| \cdot \|g(n)\|} = h(m - n), \quad (6)$$

where $Cos$ and $<>$ denotes cosine similarity and inner product, $h$ is a function of $m - n$. For simplicity, we assume $g(m)$ and $g(n)$ are two-dimension vectors. Thus, $< g(m), g(n) >$ can be written as $\Re[g(m)g^*(n)]$ where $\Re$ symbolises the real part of a complex value. With the exponential form of a complex number, we have:

$$g(m) = \|g(m)\|e^{i\Theta(m)}, \qquad g(n) = \|g(n)\|e^{i\Theta(n)}. \quad (7)$$

Then Eq. (6) can be translated as $Cos(g(m), g(n)) = \frac{\Re[g(m)g^*(n)]}{\|g(m)\| \cdot \|g(n)\|} = \Re[e^{i(\Theta(m) - \Theta(n))}] = h(m - n)$. One of the closed-form solution of $\Theta(m)$ can be derived as $\Theta(m) = m\theta$, which further yields $g(m) = \|g(m)\|e^{i(m\theta)}$. Specifically, $e^{i(m\theta)}$ can be written as $\begin{bmatrix} cos(m\theta) \\ sin(m\theta) \end{bmatrix}$. In implementation, $\|g(m)\|$ can be approximated by a grid or absorbed by a network, $\theta$ is pre-defined or learned parameter. The 2-dimension case of $g(m)$ can be easily generalized to higher dimension by stacking multiple 2-dimension cases, whose form coincides with the heuristic position encoding adopted in

| Regularizations | Continuous Gauge Transformation | | | | Discrete Gauge Transformation | | | |
|---|---|---|---|---|---|---|---|---|
| | S-NeRF | S-NSVF | DTU | T&T | S-NeRF | S-NSVF | DTU | T&T |
| **Baseline** (w/o Regularization) | 0.021 | 0.017 | 0.014 | 0.019 | 0.102 | 0.094 | 0.063 | 0.078 |
| **+StruReg** (Tretschk et al., 2021) | 0.372 | 0.360 | 0.325 | 0.354 | N/A | N/A | N/A | N/A |
| **+CycleReg** (Xiang et al., 2021) | 0.547 | 0.562 | 0.568 | 0.522 | 0.105 | 0.086 | 0.076 | 0.088 |
| **+InfoReg** | **0.793** | **0.752** | **0.772** | **0.806** | **0.992** | **1.000** | **0.996** | **1.000** |

Table 1: The PSNR performance with different regularizations for the learning of continuous gauge transformation (3D space → 2D plane) and discrete gauge transformation (3D space → 256 vectors).

| Gauge Transformations | S-NeRF | | S-NVSF | | DTU | | T&T | |
|---|---|---|---|---|---|---|---|---|
| | PSNR↑ | SSIM↑ | PSNR↑ | SSIM↑ | PSNR↑ | SSIM↑ | PSNR↑ | SSIM↑ |
| **Orthogonal Mapping** ⋄ | 18.20 | 0.816 | 19.32 | 0.833 | 23.31 | 0.845 | 19.97 | 0.829 |
| **Learning Continuous Gauge** ⋄ | **30.74** | **0.919** | **32.51** | **0.949** | **27.06** | **0.873** | **25.95** | **0.867** |
| **Spatial Hash Function** ⋆ | 27.06 | 0.902 | 29.81 | 0.932 | 27.02 | 0.876 | 26.22 | 0.869 |
| **Learning Discrete Gauge** ⋆ | 26.56 | 0.893 | 29.13 | 0.925 | 26.21 | 0.863 | 24.79 | 0.855 |
| **Learning Top-4 Gauge** ⋆ | **29.39** | **0.917** | **31.39** | **0.942** | **28.77** | **0.887** | **27.14** | **0.892** |

Table 2: Performance of different methods (pre-defined or learned) for gauge transformations. ⋄ and ⋆ indicate continuous gauge transformation (3D space → 2D plane) and discrete gauge transformation (3D space → 256 vectors). Orthogonal mapping and spatial hash function are typical pre-defined functions for continuous and discrete gauge transformations, respectively.

NeRF (Mildenhall et al., 2021). The yielded parameters $\|g(m)\|$ and $\begin{bmatrix} cos(m\theta) \\ sin(m\theta) \end{bmatrix}$ can be leveraged to index neural fields via MLP networks or feature grids.

### 3.4 DISCUSSION

As a general framework, learning gauge transformations with InfoReg or InfoInv gauge transformation can be applied in various computer vision or computer graphic tasks.

As concrete examples, learning gauge transformation with InfoReg can be applied to 3D space → 2D plane mapping to learn explicit and editable textures (Xiang et al., 2021), or 3D space → codebook mapping to yield compressed neural fields (Takikawa et al., 2022). However, with relatively high computation cost, learning gauge transformation with InfoReg cannot be applied to fast scene representation or reconstruction naively. We leave this efficient extension as our future work.

As a kind of gauge transformation, InfoInv can be applied to index neural fields directly with multi-scale setting (depending on how many different $\theta$ are selected). Notably, InfoInv allows to preserve scene information inherently without the cumbersome of regularization, which means it can be seamlessly combined with other gauge transformations (e.g., triplane projection and spatial hash function) to achieve fast scene representation and promote rendering performance.

## 4 EXPERIMENTS

### 4.1 IMPLEMENTATION

The gauge network $\mathcal{M}$ for learning gauge transformations can be a MLP network or a transformation matrix, depending on the specific downstream application. For the mapping from 3D space to 2D plane to get (view-dependent) textures, the neural field is modeled by a MLP-based network which takes predicted 2D coordinates, i.e., output of the gauge network, and a certain view to predict color and density. For the mapping from 3D space to discrete codebooks, the neural field is modeled by looking up features from the codebook, followed by a small MLP of two layers to predict color and density. By default, the codebook has two layers and each layer contains 256 vectors with 128 dimensions. In line with Instant-NGP (Müller et al., 2022), the 3D space is also divided into two-level 3D grids with size $16 \times 16 \times 16$ and $32 \times 32 \times 32$ for discrete gauge transformation.

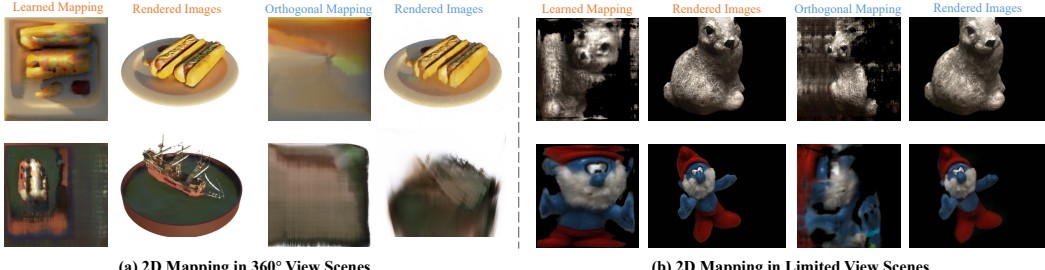

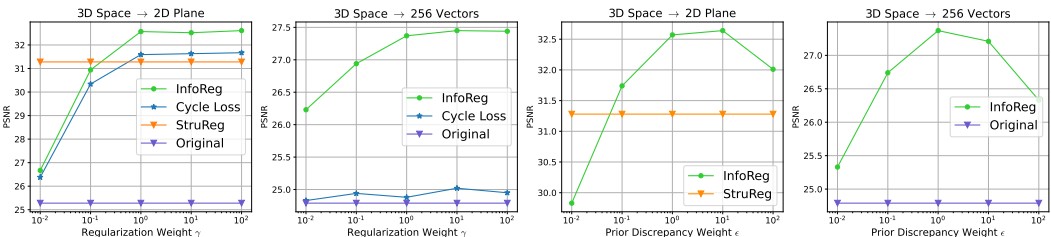

Figure 5: Qualitative results of continuous gauge transformation from 3D space to 2D plane on (a) 360° view scenes and (b) limited view scenes.

Figure 6: Ablation study of the regularization weights and prior distribution discrepancy weight.

## 4.2 EVALUATION METRICS

The introduced neural gauge fields focus on learning gauge transformation, the experiments are therefore performed to evaluate the performance of gauge transformations. By default, the experiments are conducted on continuous gauge transformation from 3D space to 2D plane and discrete gauge transformation from 3D space to 256 codebook vectors. Specifically, the effectiveness of regularization can be evaluated by measuring the *space occupancy* in the target gauge space, which refers to the percentage of mapped area in continuous cases or the utilization ratio of codebook in discrete cases. Besides, as a gauge transformation can be applied to index a radiance field, the gauge transformation can be evaluated by the novel view synthesis performance of a radiance field.

## 4.3 EVALUATION RESULTS

**Regularization Performance.** We conduct the comparison of different regularization methods including structural regularization (StruReg), Cycle regularization (CycleReg), and our InfoReg. As tabulated in Table 1, the proposed InfoReg outperforms other regularization clearly in both continuous and discrete cases. Particularly, StruReg and CycleReg can bring certain improvements for continuous cases, while StruReg can not be applied to discrete cases and CycleReg only brings marginal gain for discrete cases.

**Pre-defined vs Learned Mapping.** As shown in Table 2, we compare the performance of the pre-defined functions and the learned mapping for gauge transformation. For continuous cases, the learned mapping with InfoReg outperforms the pre-defined orthogonal mapping clearly. For discrete cases, the standard learned discrete mapping (top-1 gauge) is slightly inferior to the pre-defined spatial hash function, while the top-4 variant of the learned discrete mapping surpasses the pre-defined function substantially.

**Qualitative Results.** To demonstrate the superiority of learned gauge transformation over pre-defined function (i.e., orthogonal mapping), we visualize the mapping results from 3D space to 2D plane in Fig. 5. Fig. 5 (a) illustrates the mapping on the challenging 360° view scenes. The learned mapping can still well capture the scene texture, while the orthogonal mapping presents poor mapping performance and even fails in certain case. Fig. 5 (b) compares the mapping on limited view scenes. The orthogonal mapping is still prone to lose much texture information thanks to the inherent inflexibility.

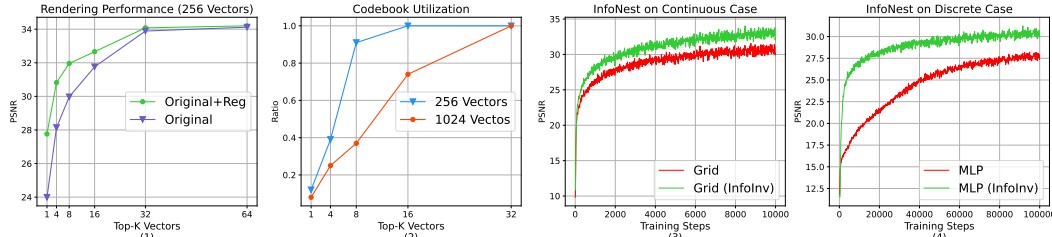

Figure 7: The left two figures: rendering performance and codebook utilization with different k values in top-k gauges. The right two figures: the performance gain with the inclusion of InfoInv.

| Models | Model Training Details | | Synthetic-NeRF | | Synthetic-NSVF | |
|---|---|---|---|---|---|---|
| | Steps↓ | Time(s)↓ | PSNR↑ | SSIM↑ | PSNR↑ | SSIM↑ |
| **TensoRF** | 30k | 708 | 32.32 | 0.958 | 35.32 | 0.976 |
| **TensoRF+InfoInv** | 30k | 721 | **32.96** | **0.962** | **35.81** | **0.979** |
| **Instant-NGP** | 30k | 458 | 32.43 | 0.960 | 35.43 | 0.977 |
| **Instant-NGP+InfoInv** | 30k | 496 | **32.91** | **0.962** | **36.01** | **0.981** |

Table 3: The rendering performance gain by combining our information-invariant gauge transformation to SOTA methods on Synthetic-NeRF and Synthetic-NSVF. Notably, only very slight computation and memory cost are introduced with the including of neural gauge fields.

**Top-k Ablation.** As defined in Eq. (1), different k can be selected for discrete gauge transformation. Figs. 7 (1) and (2) ablate the rendering performance and codebook utilization with different $k$ values. The rendering performance and codebook utilization both are improved consistently with the increase of $k$, which means the gauge collapse is gradually alleviated. The performance gain brought by InfoReg is also narrowed with the increase of $k$. We conjecture that a larger k leads to higher information capacity which alleviates the information loss in gauge transformation.

**InfoReg Parameter Study.** Fig. 6 examines the effect of the regularization weight $\gamma$ and prior discrepancy weight $\epsilon$ in Eq. (3) on the Lego scene. With the increase of regularization weight, the regularization performance of InfoReg is improved consistently, although it is saturated with a large weight. The regularization performance presents a positive correlation with a small prior discrepancy weight and a negative correlation with a large prior discrepancy weight.

**InfoInv Performance.** To demonstrate the power of information-invariant gauge transformation, we apply InfoInv to both grid-based and MLP-based neural fields for novel view synthesis. As shown in Figs. 7 (3) and (4), both convergence speed and final performance of triplane projection and spatial hash function are improved with the including of InfoInv. Besides, we also combine our InfoInv with SOTA methods including TensoRF and Instant-NGP. As shown in Table 3, the inclusion of InfoInv boost SOTA methods consistently with preferred model size. Besides, we also combine InfoInv with widely adopted transformations including triplane projection and spatial hash function. Specifically, $\|g(m)\|$ is approximated by the indexed feature in original TensoRF or Instant-NGP, $cos(m\theta)$ and $sin(m\theta)$ are used to index an auxiliary grid. As shown in Table 3, InfoInv benefits these SOTA methods consistently with ignorable (or very slight) increase of training time and model size.

## 5 CONCLUSION

This work introduces a general framework of neural gauge fields for unifying various gauge transformations in neural fields. With a target to optimize the gauge transformation along with neural fields, we establish the learning paradigm for continuous and discrete gauge transformations and derive a regularization mechanism to solve the issue of gauge collapse from the perspective of information preservation. As optimizing gauge transformation with regularization usually incurs high computation cost, a nested gauge transformation is introduced to perform instant transformation and preserve scene information with negligible computation cost. The extensive experiments and thorough analysis show that the derived gauge transformation methods outperform their vanilla counterparts, which demonstrates the potential of gauge transformations in neural fields.

## 6    ETHICS STATEMENT

Our neural gauge fields focus on technically advancing neural scene representations and are not biased towards specific race, gender, or region. As a general framework, neural gauge fields can be applied to various fields, e.g., virtual reality, artistic creation, and game design. On the other hand, the achievements in the image synthesis quality lurk the risk of being misused, e.g., illegal impersonation of actors, or face synthesis for malicious purposes. Therefore, it is also of importance to explore safeguarding technologies that can reduce the potential for misuse. One potential way is to use our approach as well as other image synthesis approaches as a discriminator to identify the synthesized images automatically. This has been actively explored by the community, e.g. Mirsky & Lee (2021). Alternatively, one should explore watermarking technologies to identify synthesized images. Besides, access control should also be carefully considered when deploying the relevant techniques to alleviate the risk of misuse. Finally, we believe creating appropriate regulations and laws is essential to solving the issue of technology misuse while promoting its positive effects on technology development.

## 7    REPRODUCIBILITY STATEMENT

We attach the source code of neural gauge fields in the supplementary material. Additionally, we provide detailed implementations in the manuscript that enable a full reproduction of the experimental results. For example, we specify the choices of rendering resolution, model architecture, and the codebook setting for discrete gauge transformation. All datasets used in our experiments are publicly accessible. Finally, all pre-trained models will be made publicly available upon the publication of this work. The implementations will enable researchers to reproduce the results as well as extend this work to future research.

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

# A APPENDIX

## A.1 GAUGE TRANSFORMATION BACKGROUND

In normal usage, gauge means a measuring instrument or measuring system, e.g., pressure gauge and temperature gauge. The transformation between different gauges (e.g., Fahrenheit and Celsius temperature) can be termed as **Gauge Transformation**. In physics, a gauge is a particular choice or specification of vector and scalar potentials which will generate a physical field. A transformation from one physical field configuration to another is called a gauge transformation. The gauge transformations for classical general relativity Wald (2010) are arbitrary coordinate transformations.

Under the context of neural fields, the measuring systems (i.e., gauge) of the 3D scene can be extended to more general definition, e.g., 3D Cartesian coordinate system, 2D coordinate system, hash table space, and frequency space. The transformation between different scene measuring system is dubbed as **Gauge Transformation**, and the neural fields with certain gauge transformation is termed as **Neural Gauge Fields** as shown in Fig. 8.

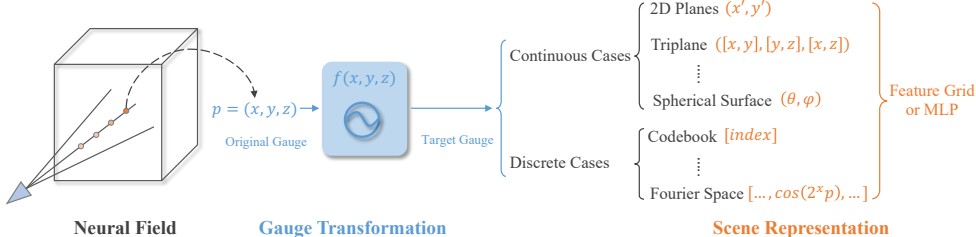

Figure 8: The framework of neural gauge fields which consists of a neural field and a gauge transformation. The gauge transformation maps original 3D coordinate to another measuring system (e.g., 2D plane and codebook) for scene representation via feature grid or a MLP network.

## A.2 DERIVATION OF INFOREG

According to Eq. (2), the final optimization loss can be formulated as:

$$\mathcal{L}_{reg} = \min_{p(y|x)} \left\{ -\gamma \iint p(y|x)p(x) \log \frac{p(y|x)}{p(y)} dxdy + \epsilon \int p(y) \log \frac{p(y)}{q(y)} dy \right\}, \quad (8)$$

where $\gamma$ and $\epsilon$ denote the weight of regularization and prior distribution discrepancy. Then, we have:

$$
\begin{aligned}
L_{reg} &= \min_{p(y|x)} \left\{ -\gamma \iint p(y|x)p(x) \log \frac{p(y|x)}{p(y)} dxdy + \epsilon \iint p(x|y)p(y) \log \frac{p(y)p(y|x)}{q(y)p(y|x)} dxdy \right\} \\
&= \min_{p(y|x)} \left\{ \iint p(y|x)p(x) \left[ -(\gamma + \epsilon) \log \frac{p(y|x)}{p(y)} + \epsilon \log \frac{p(y|x)}{q(y)} \right] dxdy \right\} \\
&= \min_{p(y|x)} \left\{ -(\gamma + \epsilon) \cdot I(X, Y) + \epsilon \cdot \mathbb{E}[KL(p(y|x)||q(y))] \right\}
\end{aligned}
$$

$$(9)$$

For $I(X, Y)$, we have:

$$I(X, Y) = \iint p(y|x)p(x) \log \frac{p(y|x)p(x)}{p(y)p(x)} = KL(p(y|x)p(x)||p(y)p(x)) \quad (10)$$

Thus, we can get:

$$\mathcal{L}_{reg} = \min_{p(y|x)} \left\{ -(\gamma + \epsilon) \cdot \mathbb{E}[KL(p(y|x)p(x)||p(y)p(x))] + \epsilon \cdot \mathbb{E}[KL(p(y|x)||q(y))] \right\}, \quad (11)$$

which can be further solved as described in the main paper.

## A.3   RADIANCE CONTRIBUTION

As 3D points near the scene surface are of more significance in mutual information, we leverage the radiance contribution weights per shading point to weight the mutual information term as in Xiang et al. (2021). Specifically, with volume density $\sigma$ and radiance $c$ of a 3D scene, the RGB color $I$ can be computed by aggregating the radiance values of shading points on the ray as below:

$$I = \sum_i T_i(1 - \exp(1 - \sigma_i \delta_i))c_i, \quad T_i = \exp(-\sum_{j=1}^{i-1} \sigma_j \delta_j), \tag{12}$$

where $i$ denotes the index of a shading point on the ray, $\sigma_i$ and $c_i$ denote the density and radiance at shading point $i$, $\delta_i$ denotes the distance between two consecutive points, $T_i$ is the transmittance. Thus, the radiance contribution weight of a shading point can be formulated as $w_i = T_i(1 - \exp(\sigma_i \delta_i))$. The $w_i$ is applied to weight the positive pairs in Eq. (5).

## A.4   DIFFERENTIABLE TOP-K

With a predicted discrete distribution $P_x = [p_1, p_2, \cdots, p_N]$, the top-k operation aims to select the k most probable index in a differentiable way so that the gradient can be back-propagated. For forward-propagation, we use the efficient top-k operation implemented in PyTorch to select the top-k values and indices. For back-propagation, we apply the reparameterization trick (Bengio et al., 2013) to approximate the gradient by replacing the top-k operation with Softmax. The detailed pseudo code of the differentiable top-k operation is given in Algorithm 2.

---

**Algorithm 2** Pseudo code of differentiable top-k operation.

---

**Input:** k values in top-k, a 3D point $x$, predicted distribution $P_x = [p_1, p_2, \cdots, p_N]$, codebook $\mathcal{V}$.
    **Forward propagation:**
        1. topk_init=$[0, 0, \cdots, 0] \in \mathbb{R}^N$
        2. index, value = Top-k$(P_x)$
        3. topk_hard = topk_init.scatter(index, value)
        4. value_sum = value.sum()
        5. topk_hard = topk_hard ÷ value_sum
        6. feature lookup: $v_x$ = Matmul(topk_hard, $\mathcal{V}$)
    **Backward propagation:**
        1. index_soft = Softmax$(P_x)$
        2. feature lookup: $v_x$ = Matmul(index_soft, $\mathcal{V}$)
**Output:** feature $v_x$.

---

## A.5   IMPLEMENTATION

We implemented our neural gauge fields and re-implemented all compared methods without customized CUDA kernels. All models are optimized for 150k steps with a batch size of 1024 pixel rays.

**Efficient Gauge Transformation.** The proposed neural gauge fields can be applied to benefit existing models without increasing their computation cost and training speed significantly. Here, we provide several strategies to achieve it: First, instead of using MLPs to learn the gauge transformation, we can directly optimize tensors for transformation, e.g., 2D offset grids to indicate the coordinate transformation in TensoRF, 1D vectors for grid points to indicate codebook index distribution in Instant-NGP. Thus, the convergence speed of gauge transformation can be aligned with grid-based models like TensoRF and Instant-NGP. Second, gauge transformation learning can be performed in the middle training stage instead of the full training process, then the learned transformation is frozen for ensuing model training. Thus, the computation cost introduced by learning gauge transformation (and regularization) can be ignorable compared with the total training time. Third, we empirically find that the prior distribution regularization term in InfoReg has been enough to avoid gauge collapse in most cases, which is much more efficient than the complete InfoReg.

## A.6 LIMITATIONS

This work focuses the fundamental problem of learning gauge transformation along with neural fields and is orthogonal to the SOTA methods for scene representation, e.g., TensoRF and Instant-NGP. Although we demonstrate that the learned gauge transformation can outperform its pre-defined counterpart (e.g., orthogonal mapping and spatial hash function), there are still some challenges to apply this method to the SOTA models.

For instance, the original Instant-NGP employs a 16-layer codebook with a size ranging from $2^{14}$ to $2^{24}$. When replacing the spatial hash function in Instant-NGP with the learned counterpart, the gauge network size will be extremely huge with an output layer dimension ranging from $2^{14}$ to $2^{24}$. One possible solution is firstly grouping the codebook into several sub-blocks with smaller size, followed by smaller gauge network to learn the mapping to each block.

## A.7 DISCUSSION & FUTURE WORK

As a general framework, the proposed neural gauge fields can be extended in several directions. Firstly, this work focuses on neural fields for novel view synthesis, which can be easily applied to downstream tasks, e.g., surface reconstruction with SDF, dynamic scenes modelling. Secondly, our InfoReg solves the gauge collapse in discrete gauge transformation, and can be potentially applied to the field of discrete representation learning (e.g., VQ-VAE (Van Den Oord et al., 2017)) which also suffers from codebook collapse. Thirdly, this work mainly explores the pure pre-defined or learned mapping, and how to combine pre-defined mapping and learned mapping to yield better performance is also of great potential. Please refer to Appendix A.6 for more discussion of limitations.

## A.8 MORE QUALITATIVE RESULTS

Fig. 9 compares the performance of InfoReg with Euclidean distance and cosine distance, the Euclidean distance is prone to preserve better texture structure compared with Cosine distance although they both cover the 2D space and achieve similar rendering performance.

Fig. 10 shows additional samples of leaning continuous gauge transformation with different regularizations.

Except for novel view synthesis, we also apply our neural gauge fields to the downstream task of texture editing as shown in Fig. 11. Specifically, we learn the mapping from 3D space (points on object surface) to sphere surface which is visualized by cubemap for texture editing.

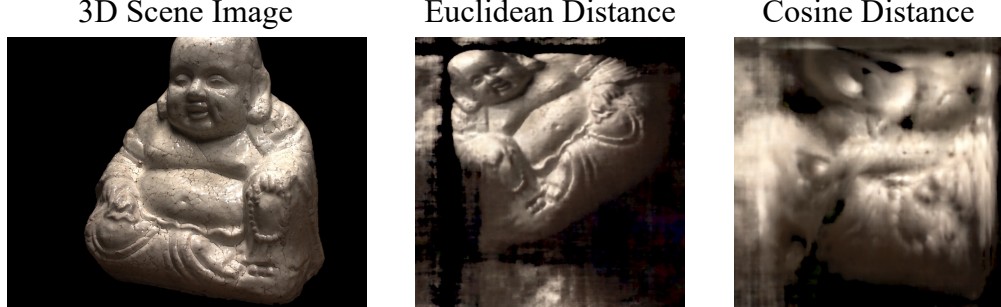

Figure 9: The results (3D space to 2D plane) of different distance metrics (Euclidean distance or Cosine distance) used in network $T$ in the proposed InfoReg. Euclidean distance leads to better preservation of local texture structure compared with Cosine distance.

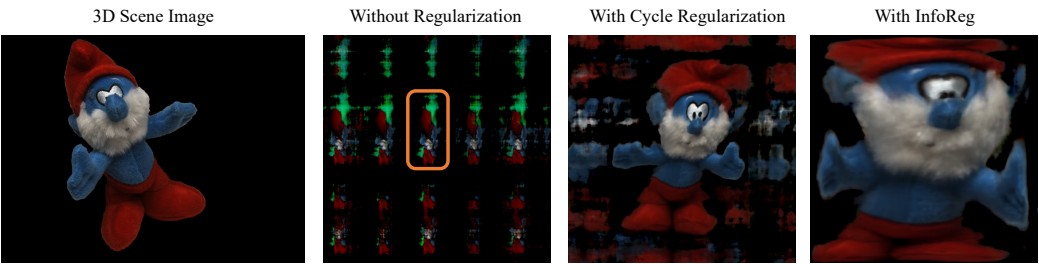

Figure 10: The results (3D space to 2D plane) of different learning regularizations. Our InfoReg leads to the best mapping results by utilizing the full plane.

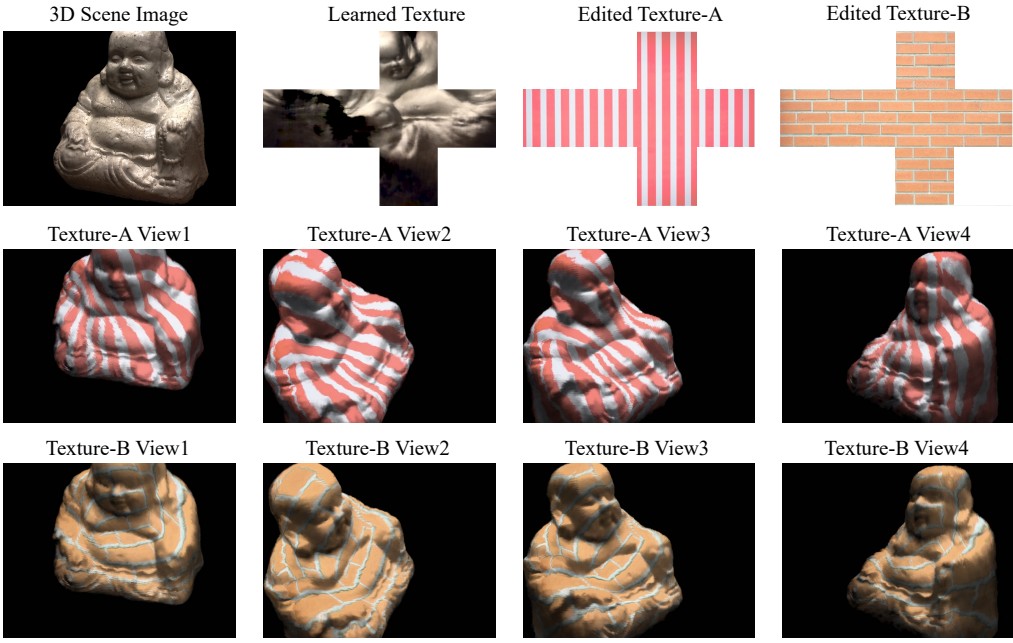

Figure 11: The application of neural gauge fields for texture editing. A gauge transformation is learned from 3D space to sphere surface (visualized by cubemap). By editing the texture of cubemap, the model is able to render multi-view images of the corresponding texture.

