# OpenReview forum: "General Neural Gauge Fields"
_ICLR.cc/2023/Conference — ICLR 2023 poster_

### Official Review · Reviewer_mQvE · 2022-10-25

**Confidence:** 2
**Correctness:** 3
**Technical Novelty And Significance:** 3
**Empirical Novelty And Significance:** 3
**Recommendation:** 5

**Clarity, Quality, Novelty And Reproducibility:**

Although I believe this work could present new ideas, I feel the presentation requires significant improvement to evaluate novelty as not an expert in the topic.

**Strength And Weaknesses:**

Strengths
-----------

Overall I believe the idea of exploring to learn a gauge transformation in an end-to-end manner is an important research direction, and this work makes contributions in this regard.

Results show better performance than state-of-the-art methods.


Weaknesses
--------------
Overall, this paper was hard to evaluate because I am not an expert in the neural radiance field and because I believe the presentation requires improvement. The elaboration on existing methodologies and new ideas is not in a way that presents the reader with an abstraction of the theory behind recent advances to then clearly and concisely show their contributions.






**Summary Of The Paper:**

This paper proposes a framework for neural gauge fields in the context of radiance fields and experimentally shows that the proposed regularisation is superior to alternative ones. This work also introduced the idea of a top-k gauge that gradually alleviates the need for regularisation.

**Summary Of The Review:**

Despite they could be a significant theoretical contribution in trying to present a new framework for gauge fields, I could not really judge the impact and novelty in the way this paper is written.

---

> ### Author Response · Authors · 2022-11-18
> **Response to Reviewer mQvE**
>
> We are sincerely thankful to every reviewer for reading our research narrowly and giving us thoughtful feedback. We carefully respond to each of the concerns and questions, together with a revised manuscript that reflects all comments.
>
> ------------------------------------------------------------------------------------------------------------------
>
> **Q1. The elaboration on existing methodologies and new ideas is not clear and concise.**
>
> We have updated our manuscripts to describe more research background of gauge transformation in the updated Appendix A.1. We also include an overview illustration of our neural gauge field framework as shown in Fig. 8.
> We hope these additional background and illustration can help the readers to understand the methodologies and our ideas.
>
> If Reviewer mQvE has further questions or suggestions for our paper, we will appreciate and are willing to make further clarification.

---

> ### Author Response · Authors · 2022-12-12
> **Thanks for your reviews!**
>
> Thank you once again for your reviews and suggestions. At the end of the discussion stage, we would like to make sure that the updated information of this paper (summarized in the general response) has been brought to your attention.
>
> We hope these additional updates \& revision and comments from other reviewers could help to solve your concerns on this work.

---

### Official Review · Reviewer_Bfkm · 2022-11-03

**Confidence:** 4
**Correctness:** 3
**Technical Novelty And Significance:** 3
**Empirical Novelty And Significance:** 2
**Recommendation:** 6

**Clarity, Quality, Novelty And Reproducibility:**

The paper is well-written and clear mostly.

Novelty: Yes as commented above

Reproducibility: not sure since the implementation section is not clear to me.

**Strength And Weaknesses:**

What's good:
1) The idea of transforming coordinate learning into a distribution regulation problem is interesting and elegant.
2) The usage of KL divergence and EM distance loss for coordinate space mapping seems novel and interesting.
3) The paper provides a detailed mathematical derivation of the relationship between mutual information maximization and the proposed KL divergence loss, but I could not read over the derivation due to this emergency review.
4) The proposed method shows better quantitative rendering scores than the previous approach.

To be improved:
1) The overall pipeline is unclear to me, a pipeline figure could really help to better understand the proposed method.
2) The paper lacks a key highlight that motivates me to use it in practice, it's good to perform better than the other two space-wrapping-based approaches, but the scores still have a big gap with SOTA and suffer from high computation cost. Would be better if the authors can figure out some unique properties in the proposed method. How does it perform for radiance field editing and rendering speed? Can it combine with the traditional rasterization rendering framework for fast rendering since the features are distributed in 2D space?
3) Table 3 is tricky to me, why not use the default configuration in the TensoRF and Instant-NGP? Seems the total parameters in Instant-NGP are significantly smaller than TensoRF under your configuration but the model size in instant-NGP is larger than TensoRF: 2^14 * 8 * 2 (even assuming 2^14 entries for each level) vs. 64 * 64 * 27 * 3, please correct me if any misunderstanding to the counting. And could be better to list the training time to have a better side-by-side comparison.
4) The reference to the convolutional occupancy field is missing, which is early work that also uses space wrapping.
5) Lacking the reference, discussion and comparison with "Variable Bitrate Neural Fields", which also learn a coordinate wrapping function.

**Summary Of The Paper:**

The paper addresses the feature index or coordinate mapping function in radiance field reconstruction as a gauge transformation problem. The core idea of the manuscript is to optimize the transformation by maximizing the mutual information and encouraging uniform coordinate distribution with an EM distance regulation term. Extensive experiments on syntheses and real object datasets demonstrate that the proposed method achieves better performance in terms of rendering quality and compactness compared to the recent SOTA fast radiance field reconstruction approaches (instant-NGP and TensoRF ) and space wrapping baed radiance field reconstruction methods (Non-Rigid Neural Radiance Fields and NeuTex).

**Summary Of The Review:**

The proposed regulation terms are novel and interesting to me, and happy to see the authors can clarify the questions listed in the "to be improved" section in the rebuttal period.

---

> ### Author Response · Authors · 2022-11-18
> **Response to Reviewer Bfkm**
>
> We are sincerely thankful to every reviewer for reading our research narrowly and giving us thoughtful feedback. We carefully respond to each of the concerns and questions, together with a revised manuscript that reflects all comments.
>
> ------------------------------------------------------------------------------------------------------------------
>
> **Q1. A pipeline figure could help to better understand the proposed method.**
>
> Thanks for your advice. We have included the illustration of the overall framework \& pipeline in Appendix Fig. 8 as suggested.
>
> **Q2. Key highlight that motivates the use in practice.**
>
> Our neural gauge fields can be combined with the SOTA scene representation methods (e.g., TensoRF, Instant-NGP) by replacing the predefined mapping functions with their learned counterparts to improve the representation capability. With proper design and training strategy, our method can improve the strong baselines of TensoRF and Instant-NGP without increasing the training cost and speed clearly. Please refer to our general response for more detail.
>
> **Q3. The performance for radiance field editing.**
>
> With neural gauge fields, we can map 3D objects to 2D UV maps as shown in Fig. 2 and Fig. 5.
> Additionally, we also conduct a supplementary experiment by mapping 3D space to a spherical surface as described in the updated Appendix A.9. By editing the mapped texture via a cubemap, we can edit the object texture as shown in Fig. 12 in the updated Appendix.
>
>
> **Q4. Seems the total parameters in Instant-NGP are significantly smaller than TensoRF.**
>
> We apologize for the unclear and misleading description of the model setting.
> Both TensoRF and Instant-NGP adopt coarse-to-fine model\& training settings in our implementation.
> Specifically, original TensoRF initializes planes of size $128\times 128$, which is gradually upsampled to $300\times 300$ during training.
> Similarly, our implemented TensoRF adopts an initialized size of $64\times 64$ which is gradually upsampled to $192\times 192$.
>
> For Instant-NGP, our implementation also follows the coarse-to-fine model setting, i.e., $2^{14}$ codebook for the coarsest layer and $2^{21}$ for the finest layer with 8 layers in total.
> (Original Instant-NGP has 16 layers ranging from $2^{14}$ to $2^{24}$).
> We have updated our manuscript to clarify the model settings in Appendix A.8 (Comparison with SOTA Methods).
>
>
> **Q5. Why not use the default configuration in the TensoRF and Instant-NGP.**
>
> The original TensoRF and Instant-NGP have very large model sizes, which are at least 20 times larger than our model. The significant difference in model size will lead to rather unfair comparison with our model (Top-256), we thus reduce the size of TensoRF from [$128\times 128$ (initial), $300\times 300$ (final)] to [$64\times 64$ (initial), $192\times 192$ (final)], and reduce the size of Instant-NGP from [$2^{14}$, $2^{24}$] to [$2^{14}$, $2^{21}$], i.e. [$2^{14}$, $2^{15}$, $2^{16}$, $2^{17}$, $2^{18}$, $2^{19}$, $2^{20}$, $2^{21}$].
>
>
> **Q6. Missing the reference to 'convolutional occupancy field' and 'Variable Bitrate Neural Fields'.**
>
> Thanks for sharing these two impressive works.
> The space wrapping used in 'Convolutional Occupancy Network' is an orthogonal mapping and we have included this reference in the updated related work.
>
> 'Variable Bitrate Neural Fields' is actually a vanilla case of our discrete gauge transformation without applying any regularization, while our main contributions lie in the unification of continuous \& discrete gauge transformations, and the derivation of InfoReg to avoid learning collapse during gauge transformation. We have included the reference and discussion of this work in the updated related work.

---

> > ### Comment · Reviewer_Bfkm · 2022-11-23
> > **Additional question**
> >
> > Thanks for the detailed response!
> > Reg Q4, Q5: Could you also make a comparison by increasing your model size? Because the model size is not a big issue in TensoRF and instant-NGP, they are all less than 100Mb per scene.

---

> > > ### Author Response · Authors · 2022-11-23
> > > **Clarification to additional question**
> > >
> > > Thanks for your response and suggestion! We will conduct the suggested experiment by increasing our model size, and will report the experimental result soon.

---

> > > ### Author Response · Authors · 2022-12-01
> > > **Clarification to additional question**
> > >
> > > We would like to thank you again for your constructive reviews.
> > > As suggest, we enlarge the size of our Top-256 Gauge model by simply increasing the codebook from two layers to eight layers (each codebook contains 256 vectors with 128 dimensions), with 3D grids ranging from $8\times8\times8$, $12\times12\times12$, $16\times16\times16$, $20\times20\times20$, $24\times24\times24$, $28\times28\times28$, $32\times32\times32$, $36\times36\times36$. A three-layer MLP and a two-layer MLP are followed to predicted color and density.
> > >
> > > We compare it with TensoRF and Instant-NGP (with the default model settings in our general response) and update Table 4 as below:
> > >
> > > | Models          | Model Training Details | Synthetic-NeRF | Synthetic-NSVF |
> > > |-----------------|------------------------|----------------|----------------|
> > > |                 | Steps         Size(MB) |  PSNR    SSIM  |  PSNR    SSIM  |
> > > | TensoRF         |  150k           67.19  | 33.71    0.961 | 36.37    0.979 |
> > > | Instant-NGP     |  150k           96.84  | 33.96    0.964 | 36.58    0.981 |
> > > | Top-256 Gauge   |  150k           13.21  | 34.05    0.966 | 37.39    0.984 |

---

### Official Review · Reviewer_hK9t · 2022-11-03

**Confidence:** 3
**Clarity, Quality, Novelty And Reproducibility:** Please see the strength & weakness se…
**Correctness:** 3
**Technical Novelty And Significance:** 3
**Empirical Novelty And Significance:** 2
**Recommendation:** 6

**Strength And Weaknesses:**

Strength:
1. The paper provides a novel perspective that unifies the coordinate transform in instant-NGP, tensorRF, and triplanes under the framework of gauge transform.
2. The proposed regularization shows improvement over existing regularization methods provided by prior works.
3. The paper shows that with the novel Top-4 Gauge, the method is able to perform comparably with SOTA.

Weakness:
1. Evaluation. It's not very clear to me that the using downstream novel view synthesis is the best way to evaluate Neural Gauge Fields. I believed that it's essential to show that the proposed method doesn't lag behind SOTA performance, but it's also important to show case what this new formulation should allow. I will encourage the authors to include other applications, potentially in the field of learning texture, materials, BRDF etc.
2. Writing. In the introduction and method section, it's hard for me to follow the motivation why is it beneficial to think of instant-NGP, tensorRT in the framework of gauge transform. This is not clear to me what's the theoretical benefit (cleaner? easier to reason about different mappings?) or the empricial advantages. I would also encourage the authors to include a background section about gauge transform as it can benefit the readers without enough backrgound.
3. The authors compared to instant-NGP / tensorRT, both of these two methods are very memory efficient and fast to optimize. I wonder if Neural Gauge Field subsumes these formulation, does the same benefit carries through? It would be nice to see time profiling and memory profiling if possible.

**Summary Of The Paper:**

This paper proposes a way to learn gauge transformer for neural fields applications (especially modeling neural radiance field for novel view synthesis). The main contribution of the work is to propose a unified framework to reason about different NeRF paradigms in the language of gauge transformation and proposes a regularization to enable learning such transformation without suffering from collapsing in local minimum. Empirical results suggests that the proposed regularization indeed improve performance.

**Summary Of The Review:**

My main concern about the paper is about the evaluation. The paper mainly evaluate the performance of Neural Gauge Field using novel view synthesis, without reporting optimization time, memory, or other applications (which I believed the true use of Neural Gauge Field can lay). The secondary concern is regarding it's not clear why we need the unified language in terms of gauge transform to describe different formulation such as instang-ngp, triplanes, and/or tensorRF.


-------
Update: I decided to raise my score since the authors address a number of concerns I raised.

---

> ### Author Response · Authors · 2022-11-18
> **Response to Reviewer hK9t**
>
> We are sincerely thankful to every reviewer for reading our research narrowly and giving us thoughtful feedback. We carefully respond to each of the concerns and questions, together with a revised manuscript that reflects all comments.
>
> ------------------------------------------------------------------------------------------------------------------
>
> **Q1. The results of optimization time and memory.**
>
> With proper design of gauge transformation \& training strategy, our method enables to improve the SOTA methods with very slight increase of training time and memory cost.
> Please refer to the general response for detailed experiments \& results of optimization time and memory cost.
>
> **Q2. Evaluation on novel view synthesis and the applications in other fields.**
>
> This work is motivated by recent works of EG3D and TensoRF for novel view synthesis, which map 3D space orthogonally into 2D feature planes to index radiance fields as shown in Fig. 1. Following the task of EG3D and TensoRF, we thus mainly conduct evaluation on novel view synthesis.
> In the manuscript, we also present the application of gauge transformation from 3D space to 2D UV space (view-dependent texture mapping) in Fig. 2 and Fig. 5 to validate \& evaluate our method.
>
> To further solve the concern of reviewers, we include the application on texture editing by learning the mapping from 3D space to sphere surface as shown in the updated Appendix A.9 and Fig. 12.
>
> **Q3. The motivation of unifying instant-NGP and TensoRF in the neural gauge field framework.**
>
> Both instant-NGP and TensoRF are actually performing gauge transformations (discrete or continuous) in radiance fields with pre-defined mapping functions, e.g., orthogonal mapping (continuous) in TensoRF and EG3D, spatial hash function (discrete) in Instant-NGP. The unification helps us to understand the essence of these works, and motivates us to derive a general and fundamental solution \& regularization mechanism (e.g. our InfoReg) to learn gauge transformations, instead of falling into a special case (e.g. cycle regularization which only works for continuous gauge transformation).
>
> Besides, our proposed unification also helps us to explore the essential reason of learning collapse for both continuous and discrete gauge transformations, as analyzed in Sec 3.3.
>
> Finally, as the main motivation, this unification enables us to validate the generalization of gauge transformation across different transformation types (including continuous and discrete cases).
>
> **Q4. Including a background section about gauge transform.**
>
> Thanks for your advice. Due to the page length limitation, we include a section in the updated Appendix A.1 to introduce a gauge transformation background as suggested.

---

> > ### Comment · Reviewer_hK9t · 2022-12-10
> > **Thanks for the reply!**
> >
> > The authors answer my concern regarding the additional applications other than novel view synthesis. The edited manuscript also introduces additional clarification I request. I will raise my score.

---

> > > ### Author Response · Authors · 2022-12-12
> > > **Thanks for the response!**
> > >
> > > We are grateful for your review and the response! We are very happy to see that our revision and additional clarification \& experiments address your concerns.

---

### Official Review · Reviewer_2BXg · 2022-11-04

**Confidence:** 3
**Correctness:** 3
**Technical Novelty And Significance:** 3
**Empirical Novelty And Significance:** 3
**Recommendation:** 6

**Clarity, Quality, Novelty And Reproducibility:**

Clarity:
+ Top-K strategy helps with distribution collapse. The paper claims the Top-K strategy helps with distribution collapse. The experiment shows the strategy improves rendering quality but doesn't provide evidence that it helps with distribution collapse. Also, it will be better if the paper can use a quantitative metric to evaluate the degree of "collapse".

+ Training Time. I'm wondering what the training time of this method compares with other methods. As the EMD calculation takes a long time, will the regularization terms make the training time longer than other methods?

+ Equation 1: i_x ∈ R^1 => i_x ∈ N^1

+ Uniform Distribution Range: For the continuous case, the range of q(y) is R^2 or [0, 1]^2?

+ Other manifolds: Tha paper assumes the target distribution to be a uniform distribution. Does the method also apply to other distributions? It will be interesting if these results can be provided, to show the generalizability of the method.

Quality
+ Fair.

Novelty
+ Fair.

Reproducibility
+ Fair



**Strength And Weaknesses:**

Strength:
+ The proposed method proves to be effective and has the potential to be adopted by future "gauge transformation" frameworks.

+ The paper is well-written and easy to follow for me.

Weakness:
+ The performance of the rendering tasks at inference time sacrifices computational cost.

**Summary Of The Paper:**

In this paper, the author proposes a general framework for the neural gauge field, including continuous and discrete cases. To solve the collapse problem, the author proposes a regularization term - InfoReg. In practice, the regularization term regulates the target distribution to a uniform distribution. The author also proposes a new top-k gauge mechanism that achieves a trade-off between model collapse and computational cost. Experiments show the method outperforms other regularization methods and gauge transformation methods. The method also outperforms SOTA in rendering tasks, but with higher computation costs.

**Summary Of The Review:**

This paper proposes a general method for gauging transformation, which is intellectually interesting and could be potentially applied to other frameworks and applications. My major concern is the higher training and inference time for high-quality results.

---

> ### Author Response · Authors · 2022-11-18
> **Response to Reviewer 2BXg**
>
> We are sincerely thankful to every reviewer for reading our research narrowly and giving us thoughtful feedback. We carefully respond to each of the concerns and questions, together with a revised manuscript that reflects all comments.
>
> ------------------------------------------------------------------------------------------------------------------------------------------------------------------------
>
>
> **Q1. Evaluation of degree of 'collapse'.**
>
> For the discrete gauge transformation, the degree of 'collapse' can be approximately measured by computing the percentage of utilized codebook entries as shown in Fig. 7 (codebook utilization). Alternatively, similar to Fig. 4 (b), the 'collapse' degree can also be evaluated by measuring the KL divergence between the predicted one-hot distributions and a discrete uniform distribution, which tends to be more accurate.
>
> For the continuous gauge transformation, the degree of 'collapse' can be evaluated by the occupancy in the target plane (see Fig. 7 of the updated manuscripts). Alternatively, similar to Fig. 4 (a), the 'collapse' degree can also be evaluated by the Earth Mover's distance (EMD) between the transformed points and the uniformly sampled points.
>
> We report the KL divergence for discrete gauge transformation and EMD for continuous gauge transformation in the table below:
>
>
> | Models                    | Continuous Transform [EMD] | Discrete Transform [KL] |
> |---------------------------|----------------------------|-------------------------|
> | Without Regularization    |          0.1762            |         0.0021          |
> | With Cycle Regularization |          0.0243            |         0.0018          |
> | With InfoReg              |          0.0157            |         0.0002          |
>
>
>
> Notably, the degree of 'collapse' can also be implicitly evaluated by the novel view synthesis (NVS) performance, as the NVS performance will be inferior if the transformation collapses to a local region or a small number of codebook indices.
>
>
> **Q2. The training time of the propose method and computational cost of EMD.**
>
> With the inclusion of a gauge transformation module in existing models, the training speed will be slightly slower. However, the effect on training speed can be negligible with proper training strategies of the gauge transformation.
> Please refer to our general response for detailed description and experiment results.
> Additionally, with this training strategy, the relatively high computational cost of EMD can also be ignored during training.
>
>
> **Q3. Typo in Equation 1.**
>
> Thanks for pointing it out, we have corrected the typo in the revised manuscript.
>
> **Q4. The range of uniform distribution in continuous cases.**
>
> The range of the uniform distribution is [-1, 1], as we aim to project 3D points to a finite plane to look up features/texture as in TensoRF \& EG3D and NeuTex.
>
> **Q5. The generalization to other distributions.**
>
> As suggested, we conduct the gauge transformation with the assumption of a Gaussian distribution prior in the updated Appendix A.8 and Fig. 9.
> The Gaussian distribution also helps to solve learning collapse, although presenting slightly inferior rendering performance than uniform distribution (30.51 vs 30.74) on Synthetic-NeRF.

---

> ### Comment · Reviewer_2BXg · 2022-11-22
> **Reviewer update**
>
> Thanks to the authors for your feedback! They addressed my concerns, especially the new Table 3 in the paper. Overall, I think this paper proposes an efficient component that is useful for gauge transformation and provides interesting analytical results. My score remains the same at this point.

---

### Official Review · Reviewer_HA8S · 2022-11-04

**Confidence:** 4
**Clarity, Quality, Novelty And Reproducibility:** Please refer the strengths above.
**Correctness:** 3
**Technical Novelty And Significance:** 3
**Empirical Novelty And Significance:** 3
**Recommendation:** 6

**Details Of Ethics Concerns:**

No concerning ethical issues as far as can be seen.


**Strength And Weaknesses:**

++ The paper is generally well written and easy to follow. The presented method is intuitive and meaningful. Graphical illustrations of Fig. 1, 2 , and 4 are particularly helpful.

++ Sections 3.2 and 3.3 are well motivated and offer new theoretical insights.

The empirical discovery and analysis on top-k gauge, reported in Section 3.3 and Fig. 7 are particularly interesting.

++ The supplementary material with additional pseudo code and discussion regarding the method’s limitations is helpful.


-- Comparison with SOTA method on Table 3, although highlights the benefit in terms of model size, is limited to only two scenes. It is difficult to conclude the benefit of the proposed method over SOTA without more exhaustive experiments.

--  The claim of  “trade-off between collapse and cost” on the basis of Figure 7 (in Section 4.2.2) is somewhat misleading.  The reported PSNR for the “original” method may be poor but cannot be said to collapse. As such, the reported gap in PSNR (which is not very significant) may not generalize well across scenes.

--  Qualitative results reported in Fig 8 and 9 (in the supplementary materials) are difficult to understand.

--  The limitation on “output layer dimension ranging from 2^14 to 2^24” is indeed of a serious concern.


**Summary Of The Paper:**

 This paper studies the problem of learning the gauge transformation in an end-to-end manner within learning the neural scene representation setting.  In this context, gauge transformations in  continuous and discrete cases are learned. The developed learning paradigm, which is generic,  maps a 3D point to a continuous coordinate or a discrete index in the target gauge, respectively.
This work also introduces  the concept of Information Regularization (InfoReg) from the principle of information conservation during gauge transformation. This regularization is performed by maximizing the mutual information between gauge parameters.  The paper also claims that the proposed top-k gauge is useful in achieving a tradeoff between learning collapse and computation cost.


**Summary Of The Review:**


I generally like the addressed problem and the approach made in the paper. However, it is currently difficult to know the benefit of the proposed method. This is mainly due to the limited experimental results. I suggest authors report more exhaustive experiments highlighting the contribution empirically.

---

> ### Author Response · Authors · 2022-11-18
> **Response to Reviewer HA8S**
>
> We are sincerely thankful to every reviewer for reading our research narrowly and giving us thoughtful feedback. We carefully respond to each of the concerns and questions, together with a revised manuscript that reflects all comments.
>
> ------------------------------------------------------------------------------------------------------------------------------------------------------------------------
>
> **Q1. Limited scenes for comparison in Table 3 and the benefit of the proposed method**
>
> As suggested, we conduct experiments on more scenes, including the Synthetic-NeRF dataset and NSVF dataset, as shown in Table 3 of the updated manuscript. Note the updated experiments are performed by applying gauge transformation learning to SOTA models, which is more fair regarding the comparison of model size and training speed.
>
> As shown in Table 3 (also our general response), the proposed neural gauge fields benefit the strong baseline of TensoRF and Instant-NGP with negligible increase of computational cost and training speed. Please refer to our general response for more details.
>
> **Q2. The misleading claim of 'trade-off between collapse and cost' on the basis of Figure 7.**
>
> We would like to clarify that we inherit the name of 'collapse' from the VQ-VAE community [1,2,3,4], where the model achieves inferior / poor performance (while not crashing completely) as only a small proportion of the codebook vectors are used.
> To avoid this potential confusion of terminology, we have included the references of VQ-VAE \& codebook collapse to clarify the definition of 'collapse'.
>
> **Q3. Figures in Appendix are difficult to understand.**
>
> Fig. 10 aims to compare different distance metrics (Euclidean or Cosine distance) used in InfoReg as described in Eq. (6) and Sec 4.1 (second paragraph).
> Fig. 11 presents additional results of learning gauge transformation with different regularization terms. We have updated the figures and captions to explain this in detail.
>
> **Q4. The limitation on “output layer dimension ranging from $2^{14}$ to $2^{24}$.**
>
> We would like to clarify that we also provide a feasible solution that enables to handle this issue in the Appendix A.6 Limitations.
> Specifically, a large codebook (e.g., $2^{20}$) can be divided into a set of sub-codebooks (e.g., $2^8$ sub-codebooks with of size $2^{12}$).
> Similarly, a grid space $256\times 256\times 256$ can be divided into a set of sub-grids (512 sub-grids of size $32\times 32 \times 32$).
> Then, with our neural gauge field framework, the mapping between the sub-grids and sub-codebooks can be learned, where a **very small output layer** can be employed for mapping (i.e., $2^8$=256). For grid points within the sub-grid, the spatial hash function can be applied to achieve the mapping.
>
> Overall, the output layer size can be adjusted freely according to the selection of the sub-codebook size. We validate this solution on Instant-NGP (with $2^{23}$) in the general response (including the results of memory cost and training speed).
>
> [1] Neural discrete representation learning
>
> [2] https://machinelearning.wtf/terms/codebook-collapse/
>
> [3] vq-wav2vec: Selfsupervised learning of discrete speech representations
>
> [4] Fast decoding in sequence models using discrete latent variables

---

> > ### Comment · Reviewer_HA8S · 2022-11-30
> > **Reply to the rebuttal**
> >
> > I appreciate the authors' reply addressing my concerns. Most of my concerns are addressed.
> > The provided additional experiments and modifications add sufficient value for me to raise the overall rating of the paper.

---

> > > ### Author Response · Authors · 2022-12-01
> > > **Thank you**
> > >
> > > We are truly grateful for your constructive review and comments, which really helped us in improving our paper. We are very happy to see that our revision and additional clarification have addressed most of your concerns.

---

### Decision · Program_Chairs · 2023-01-20

**Decision:**

Accept: poster

**Justification For Why Not Higher Score:**

While I am certain about the acceptance decision, I think the paper could also be a spotlight. The primary reason for leaning towards a poster is that none of the reviewers are strongly championing the work, and that the empirical gains, while consistent, are not overwhelming and do rely on some tunable hyperparameters.

**Justification For Why Not Lower Score:**

The unifying perspective provided for parametrizing 3D neural fields via intermediate mappings should be useful and interesting to the community. Moreover, the consistent benefits of learning such a mapping as well as the use of Infogain as a regularizer in discrete and continuous settings show that these could be adapted across scenarios.

**Metareview: Summary, Strengths And Weaknesses:**

The paper provides a unifying perspective on a recent trend in learning Neural Fields (TensorRF, Instant-NGP) where a pre-defined mapping is first used to map 3D points to a different space over which a neural field is learned. This work proposes a mechanism (and best practices e.g. regularization schemes) to instead learn this mapping for both, continuous and discrete cases, and demonstrates consistent improvements by doing so. Given the empirical results and the interesting theoretical perspective in an increasingly relevant area, the AC feels this paper would benefit the community. All the reviewers (except one with low confidence) recommend acceptance, and the AC concurs.

**Note From Pc:**

if the above contains the word "oral" or "spotlight" please see: "oral" presentation means -> notable-top-5% and "spotlight" means -> notable-top-25%. As stated in our emails, we are disassociating presentation type from AC recommendations